# Influence of Fixed Orthodontic Therapy on Pharyngeal Airway Dimensions after Correction of Class-I, -II and -III Skeletal Profiles in Adolescents

**DOI:** 10.3390/ijerph18020517

**Published:** 2021-01-10

**Authors:** Yara Al Senani, Al Jouharah Al Shammery, Abeer Al Nafea, Nisreen Al Absi, Omar Al Kadhi, Deema Al-Shammery

**Affiliations:** 1College of Dentistry, Riyadh Elm University, Riyadh 11564, Saudi Arabia; yara.a.alsenani@student.riyadh.edu.sa (Y.A.S.); aljouhara.h.alshammary@student.riyadh.edu.sa (A.J.A.S.); abeer.a.alnafea@student.riyadh.edu.sa (A.A.N.); nisrin.f.alabsi@student.riyadh.edu.sa (N.A.A.); 2Department of Preventive Dental Science, College of Dentistry, Riyadh Elm University, Riyadh 11564, Saudi Arabia; omar.alkadhi@riyadh.edu.sa

**Keywords:** airway management, cephalometry, orthodontic appliances, fixed, pharynx

## Abstract

The aim was to compare the influence of fixed orthodontic therapy (OT) on the pharyngeal airway space dimensions after correction of class-I, -II and -III skeletal profiles and among untreated adolescent patients. A control group comprising of untreated patients was also included. Demographics and OT-related information was retrieved from patients’ records. Measurements of airway spaces in the nasopharynx, oropharynx and hypopharynx were performed on lateral cephalograms. *p*-values under 0.05 were considered statistically significant. The results showed no statistically significant differences in the naso-, oro- and hypo-pharyngeal airway spaces among patients with class-I, -II and -III skeletal profiles and individuals in the control group. There were no statistically significant differences when naso-, oro- and hypo-pharyngeal airway spaces were compared among patients with class-I, -II and -III skeletal profiles. In conclusion, in non-extraction cases without maxillary expansion, fixed OT does not affect the naso-, oro- and hypo-pharyngeal airway spaces in patients with skeletal Class-I, -II and -III skeletal profiles. Further studies involving patients undergoing ME and premolar extraction are needed to elucidate the influence of fixed OT on the naso-, oro- and hypo-pharyngeal airway spaces.

## 1. Introduction

It is well-known that a physiological respiratory pattern and patency in pharyngeal airway space (PAS) are critical aspects that influence growth and development among children and adolescents [1,2,3]. Chambi-Rocha et al. [3] assessed the cephalometric differences in craniofacial structures among children and adolescents with nasal and oral breathing patterns. The results showed that constrictions in the PAS was associated with a greater palate length and facial skeletal conditions, such as higher vertical dimension in the lower anterior face in adolescents [3]. In this context, assessment of PAS continues to be a subject of interest in craniofacial and orthodontic research. Results by Alhammadi et al. [4] showed that skeletal class-II with long face significantly increased palatopharyngeal and glossopharyngeal airway volumes; however, the influence of fixed orthodontic therapy (OT) for the treatment of this form of skeletal anomaly remained unaddressed in this study [4].

Studies [5,6] have investigated the association between craniofacial-based therapies such as orthodontic therapy (OT) and orthognathic surgical interventions on the PAS. However, the results remain perpetually debatable [5,6]. Moreover, results from a meta-analysis showed that adjunct orthodontic treatments, such as rapid maxillary expansion, do not affect the airway below the palatal plane [7]. Based on the previous literate [5,6,7,8], it is evident that there is a lack of consensus regarding the influence of fixed OT on PAS dimensions following the correction of skeletal class-II and -III malocclusion. 

**Hypothesis** **1.***The present study was based on the hypothesis that there is no difference in the PAS among controls and adolescents undergoing fixed OT for the treatment of skeletal class-I, -II and -III malocclusion*.

The aim of the present study was to compare the influence of fixed OT on the PAS dimensions after correction of class-I, -II and -III skeletal profiles and among untreated adolescent patients.

## 2. Materials and Methods

### 2.1. Ethical Statement

All subjects gave their informed consent for inclusion before they participated in the study. The study was conducted in accordance with the Declaration of Helsinki, and the protocol was approved by the Ethics Committee of Riyadh Elm University, Riyadh Saudi Arabia (RC/IRB/2018/1331). The parents and participants were sent written information that described the objectives and methodology of the study in simple English and Arabic. Written consent was provided by the parents of all participants to have their offspring’s images to be retrospectively assessed. The parents and participants were assured that no personal information (such as photographs, name, address and/or contact details) would be used for future publication. Parents that agreed to have their offspring’s dental records evaluated for the present study were requested to read and sign an informed consent form.

### 2.2. Study Design and Grouping

The present cross-sectional study had a retrospective observational study design. Cephalograms from adolescents with class-I, -II and -III skeletal profiles and untreated adolescents with normal dental and skeletal occlusion (controls) were randomly assessed in the present investigation.

### 2.3. Eligibility Criteria 

The inclusion criteria were as follows: (a) Adolescent individuals (11 to 18 years) [9]; (b) patients who had undergone fixed OT in both arches; (c) patients with class-I (ANB 2°-4°), -II (ANB >4°), and -III (ANB <1°) skeletal patterns [10,11,12]; (d) self-reported medically healthy participants (such as patients without medical disease including diabetes mellitus and obesity) [13,14]; (e) presence of pre- and post-treatment lateral cephalograms with adequate diagnostic quality; and (f) untreated patients with skeletal class-I, -II and -III facial profiles (control group). The exclusion criteria encompassed the following: (a) Previous history of OT; (b) pathologies/morphological anomalies affecting the craniofacial structures; (c) abnormal tongue mobility and tongue thrust; (d) previous diagnosis of obstructive sleep apnea (OSA), snoring and/or pharyngeal anomalies; (e) pretreatment extractions; (f) maxillary expansion; and (g) pre-treatment extraction of teeth.

### 2.4. Demographic Data

Information about age and gender was retrieved from the patients’ dental records.

### 2.5. Fixed Orthodontic Therapy

Orthodontic records were assessed to confirm that the patients had undergone fixed OT using appliances such as fixed braces and arch wires. 

### 2.6. Study Sample (Cephalometric Analysis)

Lateral cephalometric radiographs were taken for individuals in the control group (individuals with skeletal class-I, -II and -III malocclusion in whom fixed OT had not initiated); whereas data was retrieved from dental records for patients with skeletal class-II and -III deformities. The cephalograms were taken by one clinician (YA; Kappa score 0.88) using matrix dimensions of 1840 × 1360 pixels and an image field of 195 × 263 mm. The magnification was set at 1:1.14. The X-ray tube was positioned at 152.4 cm from the target and the distance of the participant’s median plane to the film was set at 18 cm. Before the acquisition of cephalograms, all participants were trained through a series of 5 to 8 exercises to reach a relaxed position of the tongue as described in the study by Daraze et al. [15] These cephalograms were assessed using a computer-based software program (OnyxCeph^3TM^, Image Instruments GmbH Olbernhauer Str. 5 D 09125, Chemnitz, Germany). Digital tracing was performed by a trained and calibrated investigator (Kappa 0.92) to determine the sample’s skeletal patterns by measuring the A point/nasion/B point (ANB) angle. Airways spaces in the nasopharynx, oropharynx, and hypopharynx were measured using the McNamara linear method by a calibrated and experienced investigator (Kappa 0.92) who was blinded to the objectives of the present study [16]. In patients with skeletal class-I, -II and -III profiles, a lateral cephalometric radiograph was taken at baseline and at the end of OT. The time lapse between the first and second lateral cephalometric radiographs was 1.5 ± 0.2 years.

### 2.7. Cephalometric Measurements

Measurements for the airway spaces in the nasopharynx, oropharynx and hypopharynx were defined as described in a previous study [15]. In summary, airway spaces in the naso-, oro- and hypopharynx were represented as follows: (a) Nasopharynx: Distance between the posterior pharyngeal wall (PPW) and the dorsum of the soft palate; (b) oropharynx: Distance between the PPW and the tip of the uvula; and (c) hypopharynx: Distance between the PPW and the intersection between the posterior tongue contour and the angle of the mandible (Figure 1).

### 2.8. Statistical Analysis

Statistical analysis was performed using statistical software (IBM SPSS Statistics, Armonk, NY, USA). Group comparisons were performed using the analysis of variance (ANOVA) test. For multiple comparisons, the Bonferroni post-hoc adjustment test was performed. A *p*-value under 0.05 was nominated as the indicator of statistical significance. Data from previous studies [17,18] were used for sample size estimation. It was estimated that inclusion of at least a sample size of 28 individuals per group for the four groups would give a power of 80% to the study with an alpha of 0.05. 

## 3. Results

### 3.1. Demographic Data

A total of 174 pre- and post-treatment cephalograms (58, 58 and 58 from individuals with skeletal class-1, -II and -III skeletal profiles, respectively) were assessed. From the control group, 29 cephalograms were assessed. There was no statistically significant difference in the number of males and females among participants in all groups. The mean age of individuals in the control group and patients with skeletal class-I, -II and -III facial profiles was 15.6 ± 0.6, 15.6 ± 0.7, 15.3 ± 0.5, and 15.8 ± 0.5 years, respectively. There was no statistically significant difference in mean age of males and females in all groups (Table 1). 

### 3.2. Airway Dimensions in Relation to Skeletal Profile

The results showed no statistically significant differences in the naso-, oro- and hypo-pharyngeal airway spaces among patients with class-I, -II and -III skeletal profiles and individuals in the control group. There were no statistically significant differences when naso-, oro- and hypo-pharyngeal airway spaces were compared among patients with class-I, -II and -III skeletal profiles (Table 2).

## 4. Discussion

In indexed literature, there is a debate concerning the impact of fixed OT on pharyngeal airway dimensions [19]. Some authors report that fixed OT [20] reduces the pharyngeal airway dimensions, whereas others [21] stated that it decrease pharyngeal airway volume. Moreover, results from another study [5] reported that there is no differences in the airway spaces before and after OT of skeletal malocclusions. It is; however, pertinent to mention that, in previous studies [5,20,21,22], PAS was assessed in patients undergoing OT with reference to extraction of premolars and/or maxillary expansion. The factor that makes the current investigation different from previous investigations [5,20,21,22] is that patients that underwent extractions and/or ME were not sought. In addition, investigators of the present study included a group of control individuals that had never undergone any form of OT and exhibited normal dental and skeletal relationships. In this regard, the authors of the present study hypothesized that there is no difference in the PAS among controls and adolescents undergoing fixed OT for the treatment of skeletal class-I, -II and -III malocclusion. The current results are in agreement with the proposed hypothesis, as no significant difference in the PAS was observed in all groups. The authors support the results reported in the study by Stefanovic et al. [19] and Valiathan et al. [23], which showed that fixed OT with or without premolar extractions does not influence pharyngeal airway dimensions in growing patients. One justification for the insignificant change in PAS dimension, as observed in the current study, is that continuous growth and development accompanied with fixed OT modulates the PAS in a non-extraction scenario, thereby leading to comparable PAS before and after OT. Nevertheless, the present results are in contradiction to those reported in a pilot investigation by Al-Jewair et al. [24], according to which patients with the skeletal class-II profile demonstrate a decrease in PAS. The authors of the present study perceive that results by Al-Jewair et al. [24] should be interpreted with caution. It is worthwhile to consider that results by Al-Jewair et al. [24] were based on a pilot investigation. In other words, no prior sample size estimation was done by Al-Jewair et al. [24]. It is; therefore, suggested that *p*-values of investigations based on non-power-adjusted studies should be cautiously interpreted. Furthermore, the type of OT performed in the study by Al-Jewair et al. [24] was based on clear aligner therapy, whereas in the present investigation fixed OT was used for correction of skeletal discrepancies. 

In the present study, stringent eligibility criteria were imposed. For instance, patients undergoing OT with adjuvant expansion of maxilla were excluded. It has been reported that skeletal ME increases the nasal airway volume [25]. In a recent study, Liu et al. [26] investigated changes in maxillary width and nasopharyngeal spaces in young adults rapid ME combined with fixed OT. The results showed that widening of transverse maxillary deficiency after rapid ME as an adjuvant to fixed OT increases the nasopharyngeal air space [26]. Moreover, rapid ME and fixed OT were surgically assisted by Liu et al. [26]. There is a likelihood that if surgical-assisted fixed OT was performed in the patient population assessed in the current investigation with adjunct rapid ME, a difference in PAS may have been observed. This warrants additional well-designed and power-adjusted studies.

It is known that dental crowding and mandibular plane angle (MPA) are essential parameters that assess the skeletal class-I, -II and -III facial profiles [27,28,29,30]. Moreover, a high MPA also influences the airway volume in susceptible populations [31], and an increase in the upper airway dimension has been reported in patients with class-I crowding [32]. In the present study, dental crowding in the MPA was not assessed. Therefore, there is a possibility that these factors could have influenced the reported results. Another limitation is that two-dimensional lateral cephalograms were used for the assessment of airway dimensions. The cone beam computed tomography (CBCT) imaging technology facilitates three-dimensional evaluation of anatomical entities including root angulation and morphology, maxillary expansion and transverse maxillary dimensions, craniofacial anatomy in disease and health, alveolar bone fenestrations and dehiscence, vertical malocclusion and OSA, airway dimensions, temporary anchorage devices and the temporomandibular joint and related tissues [33]. The CBCT technology also facilitates imaging using well-defined reference points [34]. However, it is worth mentioning that patients undergoing CBCT analysis are exposed to radiation to a significantly greater extent in contrast to conventional two-dimensional lateral cephalograms. In this regard, routine use of CBCT-based imaging is demanding from a bioethical standpoint [35]. Nevertheless, two-dimensional conventional lateral cephalograms have limitations in terms of representing three-dimensional structures due to differences in magnifications, superimposition of bilateral structures and distortion [36]. Further studies using CBCT-generated lateral cephalograms are needed to assess the airway and soft tissue volumetric parameters in patients with skeletal craniofacial deformities.

## 5. Conclusions

The present cephalometry-based investigation concludes that in non-extraction cases without ME, fixed OT does not affect the naso-, oro- and hypo-pharyngeal airway spaces in patients with skeletal class-I, -II and -III skeletal profiles. Further studies involving patients undergoing ME and premolar extraction are needed to elucidate the influence of fixed OT on the naso-, oro- and hypo-pharyngeal airway spaces.

## Figures and Tables

**Figure 1 ijerph-18-00517-f001:**
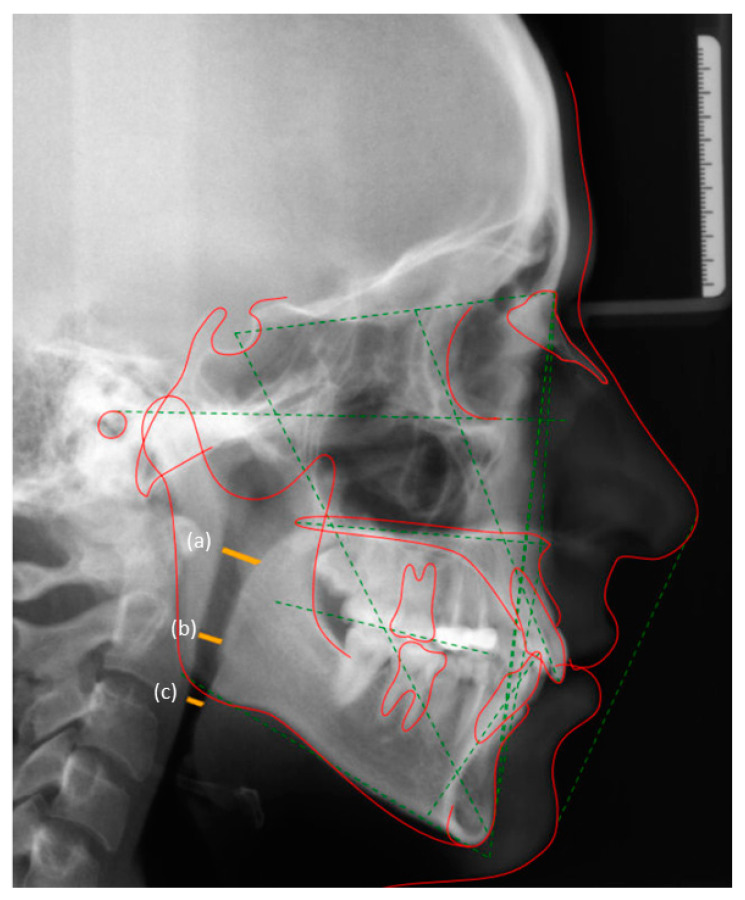
Measurement of (**a**) naso-pharyngeal, (**b**) oro-pharyngeal and (**c**) hypo-pharyngeal airway spaces on two-dimensional lateral cephalometric radiographs.

**Table 1 ijerph-18-00517-t001:** Demographics derived from patients’ dental records.

Skeletal Profiles				
Parameters	Skeletal Class-I	Skeletal Class-II	Skeletal Class-III	Control Individuals
Number of participants/lateral cephalograms (*n*)	58	58	58	29
Gender (male:female)	28:30	29:29	27:31	14:15
Overall age in years (mean ± SD)	15.6 ± 0.7 years	15.3 ± 0.5 years	15.8 ± 0.5 years	15.6 ± 0.6 years
Males	15.9 ± 0.3 years	15.2 ± 0.2 years	15.5 ± 0.4 years	15.2 ± 0.3 years
Females	15.4 ± 0.4 years	15.5 ± 0.3 years	16 ± 0.1 years	15.8 ± 0.2 years

*n* = number of individuals.

**Table 2 ijerph-18-00517-t002:** Pre- and post-treatment airway dimensions (mean ± standard deviations) among controls and patients with skeletal class-I, -II and -III profiles.

Skeletal Pattern		Airway Dimensions		
	Airway Space	Pre-Treatment (T0)	Post-Treatment (T1)	No Treatment
**Class-I**	Nasopharynx	22.51 ± 2.08	22.75 ± 1.3	NA
	Oropharynx	19.33 ± 0.95	19.35 ± 0.88	NA
	Hypopharynx	20.15 ± 1.54	20.18 ± 1.25	NA
**Class-II**	Nasopharynx	23.44 ± 2.64	24.05 ± 3.61	NA
	Oropharynx	19.46 ± 3.36	19.52 ± 3.19	NA
	Hypopharynx	20.25 ± 4.11	20.22± 4.15	NA
**Class-III**	Nasopharynx	23.58 ± 4.64	23.33 ± 4.28	NA
	Oropharynx	21.5 ± 3.17	21.61 ± 3.42	NA
	Hypopharynx	21.28 ± 4.47	21.32 ± 4.08	NA
**Control group**	Nasopharynx	NA	NA	21.32 ± 1.64
	Oropharynx	NA	NA	20.27 ± 1.16
	Hypopharynx	NA	NA	20.52 ± 3.6

NA: Not applicable. Group comparisons were done using the analysis of variance and Bonferroni post-hoc adjustment tests.

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
