# Peer review of "Influence of Fixed Orthodontic Therapy on Pharyngeal Airway Dimensions after Correction of Class-I, -II and -III Skeletal Profiles in Adolescents"

_ijerph, 2021, doi:10.3390/ijerph18020517_

Round 1

Reviewer 1 Report

Dear Authors,

Your manuscript is really interesting and provide innovative clinical results.

Please strictly follow MDPI author guidelines.

In keyword section be aware to use medical subject headings (MeSH)

In introduction section please better specify the aim of the study, following MDPI formatting style. (Hypothesis?)

"Fixed orthodontic" please specify and add a section to M&M too.

In discussion section You could evaluate to refer to these manuscripts:

1. Crimi, S.; Defila, L.; Nanni, M.; Cicciù, M.; Fiorillo, L.; Cervino, G.; Marchetti, C.; Bianchi, A. Three-Dimensional Evaluation on Cortical Bone During Orthodontic Surgical Treatment. In Journal of Craniofacial Surgery, 2020

2. https://doi.org/10.3390/jfmk4030058

Please improve conclusion section with future perspective of these article

Author Response

Reviewer 1

Dear Authors,

Your manuscript is really interesting and provide innovative clinical results.

Response from authors: Thank you for your positive feedback

Please strictly follow MDPI author guidelines.

Response from authors: The manuscript has been re-edited based on the submission guidelines of The International Journal of Environmental Research and Public Health

In keyword section be aware to use medical subject headings (MeSH)

Response from authors: The key words in the revised manuscript are MeSH terms

In introduction section please better specify the aim of the study, following MDPI formatting style. (Hypothesis?)

Response from authors: The hypothesis has been clarified and has been highlighted in the revised manuscript.

"Fixed orthodontic" please specify and add a section to M&M too.

Response from authors: As recommended, a section “Fixed orthodontic therapy” has been added to the section of Materials and Methods.

In discussion section You could evaluate to refer to these manuscripts:

  1. Crimi, S.; Defila, L.; Nanni, M.; Cicciù, M.; Fiorillo, L.; Cervino, G.; Marchetti, C.; Bianchi, A. Three-Dimensional Evaluation on Cortical Bone During Orthodontic Surgical Treatment. In Journal of Craniofacial Surgery, 2020
  2. https://doi.org/10.3390/jfmk4030058

Response from authors: The recommended references by Crimi et al (2020) and Sambataro et al (2019) have been added to the revised manuscript and the reference list has been adjusted accordingly.

Please improve conclusion section with future perspective of these article

Response from authors: The conclusion sections in the abstract as well as in the main text have been revised with emphasis on future perspectives. The revised conclusion reads as follows:

The present cephalometry based investigation concludes that in non-extraction cases without ME, fixed OT does not affect the naso-, oro- and hypo-pharyngeal airway spaces in patients with skeletal Class -I, -II and -III skeletal profiles. Further studies involving patients undergoing ME and premolar extraction are needed to elucidate the influence of fixed OT on the naso-, oro- and hypo-pharyngeal airway spaces.

Reviewer 2 Report

I have read and reviewed a previous version of the same manuscript. In the revised manuscript, the authors have answered all ym queries and have also extensively revised the manuscript. In this regard, I have no additional comments and recommend authors to revise the minor typos.

Author Response

Reviewer 2

I have read and reviewed a previous version of the same manuscript. In the revised manuscript, the authors have answered all my queries and have also extensively revised the manuscript. In this regard, I have no additional comments and recommend authors to revise the minor typos.

Response from authors: Thank you for your vigilant review and positive feedback. The manuscript has been revised by a professional scientific writer and has been correct for English vocabulary and expression. Moreover, typos have also been corrected throughout the revised manuscript.

Reviewer 3 Report

Title author affiliation: corresponding author marked as 2 and it should be 6

Introduction:

*[Studies [5, 6] have investigated the association between craniofacial-based therapies such as orthodontic therapy (OT) and orthognathic surgical interventions on the PAS in susceptible patients.]What do you refer to by “susceptible”?

*The last section of literature review is too long with a style closer to discussion rather than review

Methods

*What database was used to extract the data? Ortho clinic? What about controls? Did controls have pre and post lateral ceph? Was it taken (prospective design?) or retrieved (retrospective)?  Why were they not treated for their skeletal malocclusion? If they did not have skeletal malocclusion, then are they really an adequate control?

*Who took the airway measurements? One examiner or more? Expertise? Why wasn’t intra-and inter-examiner agreements assessed for reliability?

Stats

[it was estimated that inclusion of at least 28, 28, 28; and 28 cephalograms]

*Consider re-writing it as sample size 28 per group for the four groups….

*How did the authors compare case and control? Cases have pre and post… controls have neither… just one measurement in time but it is neither pre nor post!

*The authors list the tests they used but did not specify which test was used to compare which group nor did they clarify it in the table.

Discussion

[Al-Jewair et al. [24] according to which, patents with skeletal…]

*Spelling mistake: patients

*The discussion is long, sometimes repetitive… example:

[Another limitation is that the authors used two-dimensional imaging technology (lateral cephalograms) for the assessment of airway dimensions among individuals in the control-group and patients with skeletal class -I, -II and -III skeletal profiles.]

This can be written as: Another limitation is that two-dimensional lateral cephalograms were used for the assessment of airway dimensions.

Conclusion

It is too general. The statement should be clarified as limited to fixed OT with no extraction or expansion and using lateral ceph.

Author Response

Reviewer 3

Title author affiliation: corresponding author marked as 2 and it should be 6

Introduction:

*[Studies [5, 6] have investigated the association between craniofacial-based therapies such as orthodontic therapy (OT) and orthognathic surgical interventions on the PAS in susceptible patients.]What do you refer to by “susceptible”?

Response from authors: Thank you for watchful review and valuable comments. The sentence has been revised as follows:

Studies [5, 6] have investigated the association between craniofacial-based therapies such as orthodontic therapy (OT) and orthognathic surgical interventions on the PAS.

*The last section of literature review is too long with a style closer to discussion rather than review

Response from authors: Thank you for the comment. The last section of the literature review has been revised as follows:

Studies [5, 6] have investigated the association between craniofacial-based therapies such as orthodontic therapy (OT) and orthognathic surgical interventions on the PAS. However, the results remain perpetually debatable [5, 6]. Moreover, results from a meta-analysis showed that adjunct orthodontic treatments such as rapid maxillary expansion do not affect airway below the palatal plane [7]. Based on the previous literate [5-8],  it is evident that there is a lack of consensus regarding the influence of fixed OT on PAS dimensions following the correction of skeletal class-II and -III malocclusion.

Methods

*What database was used to extract the data? Ortho clinic? What about controls? Did controls have pre and post lateral ceph? Was it taken (prospective design?) or retrieved (retrospective)?  Why were they not treated for their skeletal malocclusion? If they did not have skeletal malocclusion, then are they really an adequate control?

Response from authors: Thank you for the query. The orthodontic treatment related data (cephalograms) was retracted from the patient’s orthodontic records. We realize that in the manuscript it was not clear how the controls were recruited; and we are pleased to provide a clarification. The control-group comprised of new patients with skeletal class -I, -II and -III malocclusions in whom fixed orthodontic therapy had not initiated. We have clarified this in the revised eligibility criteria. The revised eligibility criteria read as follows:

The inclusion criteria were as follows: (a) Adolescent individuals (11 to 18 years) [9]; (b) patients who had undergone fixed OT in both arches; (c) patients with Class -I (ANB 2°-4°), -II (ANB >4°), and -III (ANB <1°) skeletal patterns [10-12]; (d) self-reported medically healthy participants (such as patients without medical disease including diabetes mellitus and obesity) [13, 14]; (e) presence of pre- and post-treatment lateral cephalograms with adequate diagnostic quality; and (f) untreated patients with skeletal class -I, II and III facial profiles (control group).

*Who took the airway measurements? One examiner or more? Expertise? Why wasn’t intra-and inter-examiner agreements assessed for reliability?

Response from authors: Thank you for the query. The cephalometric investigations were performed by one trained and calibrated investigator. We have revised the text as follows:

Lateral cephalometric radiographs were taken for individuals in the control group (individuals with skeletal class -I, -II and -III malocclusion in whom fixed OT had not initiated); whereas data was retrieved from dental records for patients with skeletal class -II and -III deformities. The cephalograms were taken by one clinician (YA; Kappa score 0.88)

Stats

[it was estimated that inclusion of at least 28, 28, 28; and 28 cephalograms]

*Consider re-writing it as sample size 28 per group for the four groups….

Response from authors: Thank you for the recommendation. The sentence has been revised as follows:

It was estimated that inclusion of at least a sample size of 28 individuals per group for the four groups would give a power of 80% to the study with an alpha of 0.05.

*How did the authors compare case and control? Cases have pre and post… controls have neither… just one measurement in time but it is neither pre nor post!

Response from authors: Thank you for your question. The control group was included based upon the recommendation for a Reviewer when the current manuscript was previously submitted to IJERPH. Since this is a retrospective study, we assessed airway spaces on cephalograms at a pre-treatment phase and then at a post-treatment phase. This is reflected in Table 2 in the manuscript. Since the controls comprised of individuals that were not undergoing any form of dental treatment, there was no ethical and scientific reason to expose them to another cephalometric analysis.

*The authors list the tests they used but did not specify which test was used to compare which group nor did they clarify it in the table.

Response from authors: In the revised Table 2, we have clarified that group comparisons were done using the Analysis of variance and the Bonferroni Post-hoc adjustment tests.

Discussion

[Al-Jewair et al. [24] according to which, patents with skeletal…]

*Spelling mistake: patients

Response from authors: Thank you for your observation. In the revised manuscript, we have corrected the typo error. Moreover, the revised manuscript ahs been adjusted for errors pertaining to English vocabulary and expression. The revised version of the error referenced above is as follows:

Nevertheless, the present results are in contradiction to those reported in a pilot investigation by Al-Jewair et al. [24] according to which, patients with skeletal Class-II profile demonstrate a decrease in PAS.

*The discussion is long, sometimes repetitive… example:

[Another limitation is that the authors used two-dimensional imaging technology (lateral cephalograms) for the assessment of airway dimensions among individuals in the control-group and patients with skeletal class -I, -II and -III skeletal profiles.]

This can be written as: Another limitation is that two-dimensional lateral cephalograms were used for the assessment of airway dimensions.

Response from authors: Thank you for the comment. The discussion has been trimmed and references have been adjusted accordingly. The specific sentence that you referred to has been revised as recommended. The revied sentence reads as follows:

Another limitation is that two-dimensional lateral cephalograms were used for the assessment of airway dimensions.

Conclusion

It is too general. The statement should be clarified as limited to fixed OT with no extraction or expansion and using lateral ceph.

Response from authors: Thank you for the comment. The conclusion has been revised as follows:

The present cephalometry based investigation concludes that in non-extraction cases without ME, fixed OT does not affect the naso-, oro- and hypo-pharyngeal airway spaces in patients with skeletal Class -I, -II and -III skeletal profiles. Further studies involving patients undergoing ME and premolar extraction are needed to elucidate the influence of fixed OT on the naso-, oro- and hypo-pharyngeal airway spaces.
